# Application of biochar from crop straw in asphalt modification

Xinli Gan[1,2]*, Wenli Zhang[2]

**1** Research and Development Center of Transport Industry of Technologies, Materials and Equipment of Highway Construction and Maintenance, Gansu Road & Bridge Construction Group, Lanzhou, Gansu, P.R. China, **2** School of Transportation Engineering, Guizhou Institute of Technology, Guiyang, Guizhou, P.R. China

* ganxinli2007@126.com

**Data Availability Statement:** All relevant data are within the manuscript and its Supporting Information files.

**Funding:** This work was financially supported by the open fund of the Research and Development Center of Transport Industry of Technologies,

## Abstract

The objective of this study is to verify the feasibility of using biochar made from crop straw as a bitumen additive to improve some properties of bitumen. The differences between crop straw biochar prepared in a laboratory and commercial charcoal were investigated through scanning electron microscopy and laser particle size analyses. Furthermore, biochar-modified asphalt was prepared using the high-speed shear method, and the penetration, softening point, ductility at 15°C, and apparent viscosity of the asphalt binder with 6% biochar were measured at 120, 135, 150, 160, and 175°C. It was found that both the crop straw biochar and the commercial charcoal consist mainly of C, O, Si, and K, but the C content of crop straw biochar is slightly higher than that of commercial charcoal. The particle size of biochar is smaller than that of commercial charcoal, while the specific surface area is larger. It was determined that the addition of crop straw biochar significantly improved the high-temperature performance of asphalt, and that biochar and commercial charcoal have a similar influence on the high temperature performance of asphalt.

## 1. Introduction

The planting of crops produces a considerable amount of straw. Survey data reveal that three billion tons of crop straw are produced worldwide, and China alone produces one billion tons of crop straw every year. The most commonly used crop straw production methods are incineration and burial [1]. A statistical analysis showed that in the Chinese mainland only, the annual amount of straw burned is as high as 1.4 billion tons [2]. The burning of straw not only causes serious pollution to the surrounding air [3] but also leads to fire hazards. Additionally, the degradation period is longer, and failure to degrade straw in a timely manner affects the continued cultivation of the land [4]. Therefore, the development of an effective and sustainable way of using crop straw has become an important research and real-world application issue.

The performance of asphalt pavements inevitably degrades under the combined action of loads and the natural environment. During the high-temperature season, asphalt pavements absorb heat, so their temperature increases causing rut and other issues under the repeated

Materials and Equipments of Highway Construction and Maintenance (GLKF201816).

**Competing interests:** The authors have declared that no competing interests exist.

action of traffic loads [5]. The repeated action of traffic loads for an extended period of time causes fatigue and ultimately the cracking of asphalt pavements [6,7]. Asphalt ages under the influence of light and temperature; this reduces its cementation ability and negatively affects the durability of asphalt pavements [8]. Owing to the high cost and environmental sensitivity of asphalt pavements, researchers are focusing on improving their durability.

Studies have demonstrated that the addition of charcoal in the form of small particles to the asphalt binder can effectively improve the high-temperature stability, fatigue performance, and aging resistance of asphalt and asphalt mixtures and does not adversely affect other asphalt pavement performance indicators [9,10]. These studies used charcoal prepared from wood in the laboratory and commercial charcoal. Wang [11] showed that with the increase in charcoal content, the asphalt penetration decreased, and the softening point increased.

An effective way of using crop straw is as a source of biochar for asphalt. On the one hand, this reduces the environmental burden due to the improper treatment of these crop straws; on the other hand, it increases the economic value of crops and improves the utilization rate of agricultural products. Furthermore, the addition of biochar can improve the durability and increase the service life of asphalt pavement, reducing construction and maintenance costs over the complete life cycle of the asphalt pavement, which has significant economic and social value.

## 2. Literature review

Many countries have paid increasing attention to the utilization of crop straw. Doyle et al. [12] studied the economic benefit of the use of straw as a fuel, as raw material for papermaking, and as animal feed, compared to burning the straw in a field. Smuga-Kogut et al. [13] studied the use of buckwheat straw for energy purposes; among other benefits, they found that the production of second-generation bioethanol could enable its wider application and increase the cost-effectiveness of tillage. Shafie et al. [14] investigated the economic feasibility (i.e., operating, capital, and logistic costs) of rice straw co-firing at coal power plants in Malaysia and determined that co-firing rice straw in an existing coal power plant could reduce $CO_2$ emissions. Sun et al. [15] summarized the utilization methods of various types of crop straw, including soil incorporation, open burning, indoor incineration with energy recovery, combined electricity and energy recovery, conventional straw pulping, and a novel type of straw pulping, and analyzed the environmental impact of these methods. Nair et al. [16] studied the production of ethanol, biogas, and high-protein animal feed from wheat straw. Most of the existing utilization methods have specific requirements for straw. For example, the palatability of straw is important for preparing straw into feed. Not all types of straw are eaten by animals, and the use of straw to prepare chemical products has high cost. In this study, we produced biochar powder from straw and used it to prepare asphalt pavement. The advantage of this method is that it is simple and applicable to all straw.

Under low-oxygen conditions above 400°C, crop straw is pyrolyzed, and biochar is produced [17,18]. In recent years, various methods to prepare and utilize biochar from crop straw, e.g., to improve soil, treat sewage, and as fuel, have been explored [19–22]. Some researchers have begun to modify asphalt with charcoal powder to improve some of the performances of the asphalt binder. Zhao et al. [23] tested the rheological characteristics, rutting and fatigue performance, and ductility properties of an asphalt binder modified using biochar derived from switchgrass through different types of pyrolysis. They found that biochar could reduce the temperature susceptibility and significantly increase the rutting resistance of the asphalt binder. Renaldo et al. [24] investigated the effect of biochar on the improvement of asphalt aging susceptibility by blending one control binder (PG 64–22) and two bio-modified binders at concentrations of 3% and 6%, respectively; the rheological characteristics of the

specimens were then analyzed before and after aging. It was determined that biochar could be used to improve the aging resistance of asphalt binder. Zhen et al. [25] prepared biochar-modified asphalt via the high-speed shear method and conducted laboratory rolling thin film oven tests and 60°C dynamic viscosity tests of the biochar-modified asphalt; they found that biochar could increase the bond capability and resistance to flow deformation of the modified asphalt. Zhang et al. [26] used biochar as a modifier for a petroleum asphalt binder and compared the rheological properties of biochar-modified asphalt binders using different particle sizes and contents with one control and one flake graphite-modified asphalt binder. They found that the biochar-modified asphalts had higher high-temperature rutting resistance and better anti-aging properties than the graphite-modified asphalt, especially for the binders with the smaller and higher-content biochar particles. Wu et al. [27] investigated the complex modulus of biochar converted from straw as an alternative mineral filler in asphalt mastic. They found that the biochar results in an asphalt mastic with higher stiffness compared the conventional granite mineral filler. Wu et al. [28] evaluated the aging susceptibility of asphalt binders modified with biochar and found that the addition of biochar decreased the susceptibility towards aging of the bio-asphalt binders estimated using the rheological aging index.

## 3. Materials and methods

The asphalt binder used in this study was produced in Maoming, China, and its performance was tested according to the Standard Test Methods of Bitumen and Bituminous Mixtures for Highway Engineering (JTG E20-2011) of China [29], as shown in Table 1.

A muffle furnace was used to prepare the crop straw biochar at 700°C. Subsequently, the biochar was ground into powder particles. Finally, the biochar was added to the asphalt binder to prepare biochar-modified asphalt via high-speed cutting.

The preparation process was as follows:

Step 1: remove the soil and other sundries at the root of the straw, and clean it until there are no visible sundries;

Step 2: dry the straw naturally, and cut it into pieces of approximately 5 cm;

Step 3: put the straw in a 105°C blast drying oven, and dry it to a constant weight;

Step 4: the dried straws were placed in a muffle furnace at 450°C for 2 h. The muffle furnaces were sealed to maintain an oxygen-deficit environment, and the biochar was prepared under pyrolytic conditions. After cooling, the biochar was ground to a powder using a grinder.

Step 5: the ground powder was screened using a square sieve with a diameter of 0.075 mm. A biochar powder with particle diameters of less than 0.075 mm was used for standby.

Biochar-modified asphalt was prepared via the high-speed shear method through the following steps:

Step 1: heat the base asphalt to 150–160°C and add the prepared crop straw biochar to the melted asphalt at the desired dosage;

Step 2: insert the high-speed shearing machine into the asphalt for high-speed shearing for at least 30 min and then heat the asphalt continuously to ensure that its temperature is at least 150°C;

Step 3: place the modified asphalt in an environmental box at 120°C for 6 h to ensure that the crop straw biochar fully absorbs the asphalt.

**Table 1. Technical performance of asphalt binder.**

| Apparent viscosity (135°C)/(mPa·s) | Penetration (25°C, 100 g, 5 s)/0.1 mm | Softening point (°C) | Ductility (5 cm min$^{-1}$, 15°C)/cm | Density at 15°C/(g/cm$^3$) |
|---|---|---|---|---|
| 622.5 | 75.2 | 45.6 | 110.4 | 1.025 |

Step 4: take out the modified asphalt and place it in a clean place until its temperature is constant.

A commercially available charcoal was utilized for comparison. Using the same method, the commercial charcoal was ground into a powder, and a 0.075 mm square-holed sieve was utilized to ensure that the powder consisted of particles with a diameter smaller than 0.075 mm.

## 4. Experiments

### 4.1 Scanning electron microscopy

Two types of carbon powder were pasted on a conductive tape, and a ZEISS SUPRA 55 field emission scanning electron microscopy (SEM) system (Zeiss ultra 55) was used to characterize the biochar and commercial charcoal powders. In addition, the element content of the two samples was determined through an element analysis using an electron diffraction spectroscopy (EDS) module.

### 4.2 Laser particle size analysis

The particle size distribution of carbon powder has an important influence on its dispersion characteristics in asphalt and, in turn, on the physical shear time and performance of the modified asphalt in the production process. To determine the particle size distribution of the crop straw biochar and commercial charcoal powders, we used a Malvern Mastersizer 3000 laser particle size meter. The main technical parameters of the Malvern Mastersizer 3000 laser particle size meter are presented in Table 2.

### 4.3 Physical property test of the biochar and charcoal powders

The softening point, 15˚C ductility, and 25˚C penetration of the biochar-modified asphalt and commercial charcoal modified asphalt were measured when the biochar content was 0%, 2%, 4%, 6%, 8%, 10%, and 12% (percentage by mass of asphalt), according to the Standard Test Methods of Bitumen and Bituminous Mixtures for Highway Engineering (JTG E20-2011) of China [29]. The purpose of testing these indexes was to compare the effect of two types of carbon powder on the performance of asphalt, and to determine the optimal dosage of crop straw biochar when used for asphalt modification. In addition, to determine the mixing and compaction temperatures of the asphalt pavement with crop straw biochar, the optimum crop straw biochar was determined twice, and the apparent viscosity was measured at 120, 135, 150, 160, and 175˚C according to the Standard Test Methods of Bitumen and Bituminous Mixtures for Highway Engineering (JTG E20-2011) of China [29].

## 5. Results and discussion

### 5.1 Differences between biochar and charcoal

The SEM images of the two specimens are presented in Fig 1.

**Table 2. Main technical parameters of the Malvern Mastersizer 3000.**

| Parameter type | Parameter value |
|---|---|
| Red light source | Maximum 4MW He-Ne, 632.8nm |
| Blue light source | 10mW LED, 470nm |
| Effective focal length | 300mm |
| Range | 0.01–3500μm |

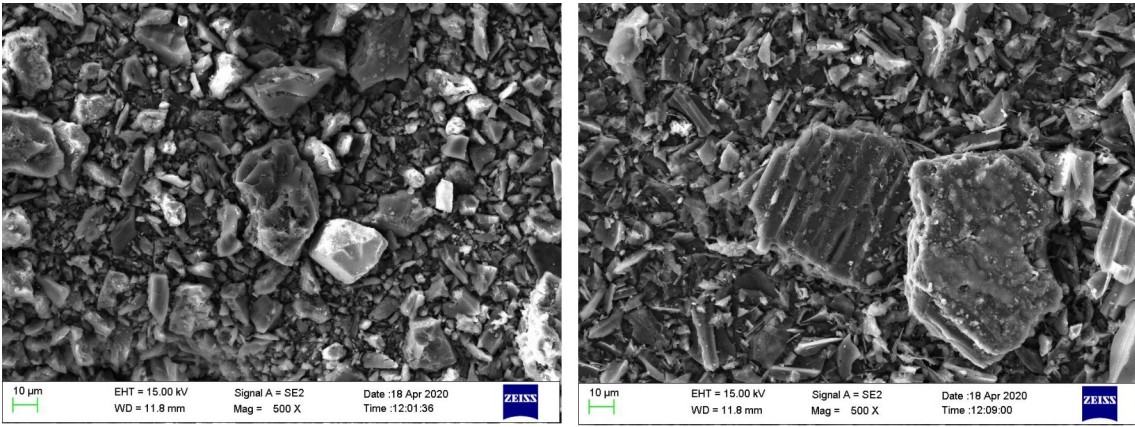

(a) Commercial charcoal powder (b) Biochar powder made from straw

**Fig 1. SEM images of the two types of carbon powder samples (500× magnification).**

As shown in Fig 1, the particle characteristics of the two types of carbon powder samples are similar, the particle shape of the powder is octahedron-like, and there is a small number of fibrous components. The mapping diagrams of the two types of test pieces are presented in Fig 2.

In Fig 2, carbon, silicon, oxygen, and potassium are marked in red, blue, green, and pink, respectively. It can be seen that both types of carbon powder mainly consist of carbon and contain some silicon, oxygen, and potassium. The element distribution of biochar is more uniform than that of commercial charcoal. Table 3 presents the results of the EDS analysis.

Table 3 and Fig 2 reveal that the two types of carbon powder contain mainly C, O, Si, and K, and the C content in the crop straw biochar is slightly higher than that in commercial charcoal. The contents of the above four elements accounts for more than 96% of the mass fraction of each test piece. In addition, small amounts of Ca and Mg were found in the two powders.

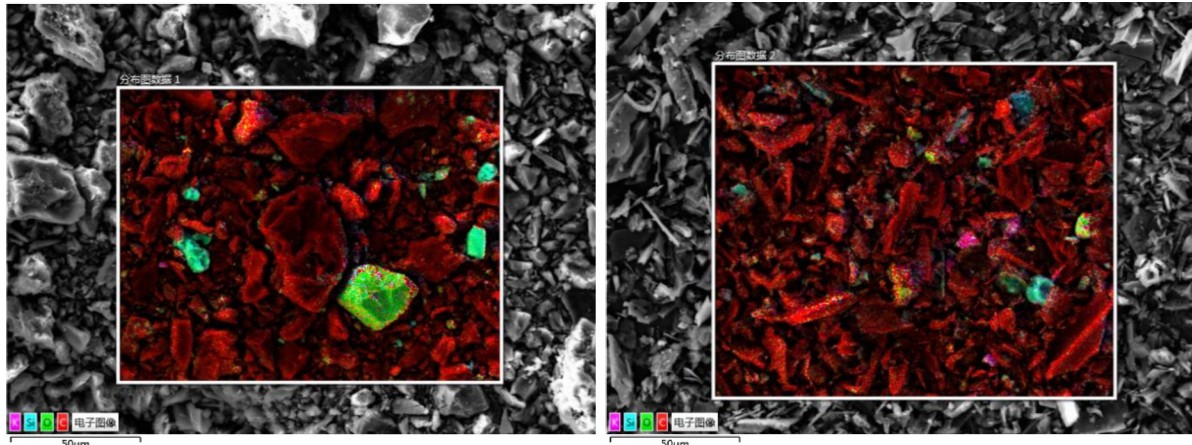

(a) Commercial charcoal powder (b) Biochar powder made from straw

**Fig 2. Mapping images of the two types of carbon powder samples.**

**Table 3. Content of each element in the two types of carbon powder.**

| Element type | Atomic percentage (%) | | Mass percentage (%) | |
|---|---|---|---|---|
| | Commercial charcoal powder | biochar powder made from straw | Commercial charcoal powder | biochar powder made from straw |
| C | 80.64 | 83.46 | 72.56 | 75.86 |
| O | 15.96 | 13.55 | 19.13 | 16.4 |
| Mg | 0.43 | 0.17 | 0.78 | 0.32 |
| Al | 0.24 | 0.00 | 0.48 | 0.00 |
| Si | 1.21 | 1.06 | 2.55 | 2.26 |
| K | 0.88 | 1.14 | 2.57 | 3.36 |
| Ca | 0.64 | 0.43 | 1.93 | 1.29 |
| Cl | 0.00 | 0.19 | 0.00 | 0.51 |

We performed a single factor ANOVA in Microsoft Excel software to analyze the differences in the mass percentage of the elements in the two kinds of charcoal. At the time of analysis, the significance level was set to 0.05; the results are presented in Table 4.

As shown in Table 4, the detection statistic F is significantly less than the critical value F crit, indicating that biochar and commercial charcoal have similar performance.

## 5.2 Particle size distribution

Because the particle size of carbon powder is very small (micron level), it is difficult to characterize it from the grading distribution diagram. Therefore, in this study, a laser particle size analysis test of the crop straw biochar and commercial charcoal was conducted. Fig 3 shows the particle volume distribution of the two types of charcoal powder.

From the figure, it can be seen that the particle size range of the crop straw biochar powder is greater than that of commercial charcoal; the particle size of crop straw biochar powder is concentrated in the range of 3–30 μm, whereas that of commercial charcoal is concentrated in the range of 5–70 μm. The grain size distribution curve of biochar was left-shifted compared to that of the commercial charcoal, which indicates a finer biochar particle size. Some biochar particles have a diameter greater than 75 μm; this is inevitable given the large number of carbon particles. Nevertheless, the amount of such large particles is very small and can be neglected.

Table 5 presents the results of the particle size analysis test of the two biochar powders. In Table 5, D[3,2] denotes the area average diameter; D[4,3] denotes the volume average diameter; D(10), D(50), and D(90) indicate that particles smaller than the listed particle size account for 10%, 50%, and 90% of the total number of particles, respectively.

Table 5 reveals that the d [3,2], d [4,3], D (10), D (50), and D (90) values of crop straw biochar are smaller than those of commercial carbon, indicating the particle size of crop straw biochar is smaller than that of commercial carbon. The commercial charcoal has a finer particle size and larger specific surface area than crop straw biochar. With equal quality, the crop straw biochar powder has a larger contact area with asphalt, so it mixes less than the

**Table 4. Results of single factor ANOVA.**

| Source of Variation | SS | df | MS | F | P-value | F crit |
|---|---|---|---|---|---|---|
| Between Groups | 1.819E-12 | 1.000 | 1.818E-12 | 2.231E-15 | 1.000 | 4.600 |
| Within Groups | 11412.269 | 14.000 | 815.162 | | | |
| Total | 11412.269 | 15.000 | | | | |

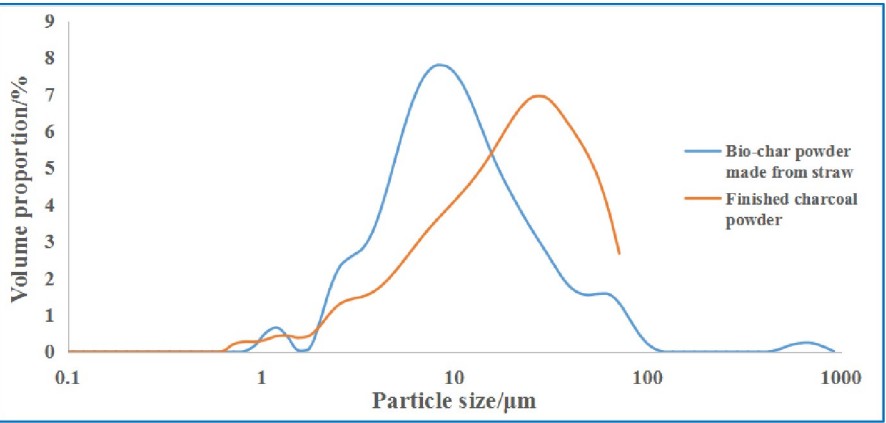

**Fig 3. Particle size distribution of the two types of carbon powder.**

commercial charcoal. Some studies have shown that smaller-sized particles (especially those smaller than 150 μm) form stronger agglomerates [30]. This is because the van der Waals force between the particles is strong when the particles are small, which induces the particles to agglomerate [31]. Owing to its smaller particle size, the crop straw biochar powder agglomerates more easily; hence, the crop straw biochar-modified asphalt requires a longer shearing time during preparation. When the powder is added into the asphalt binder, the asphalt gets wrapped on the surface of the powder particles, and interfacial tension will be induced under the combined action of the intermolecular force between the two-phase, chemical bond, and mechanical locking force. Under these effects, the plasticity of asphalt will be reduced, and deformation does not occur easily under the action of the force [32].

## 5.3 Technical performance of the modified asphalt

The technical performance test results of the crop straw biochar-modified asphalt are presented in Table 6.

It was found that, with the increase in biochar or commercial charcoal content, the asphalt penetration decreased, the softening point increased, and the ductility decreased. Many studies have shown that the softening point, penetration, and ductility of asphalt are significantly related to the high-temperature performance of asphalt [33,34]. A better high-temperature performance of asphalt leads to a higher softening point and smaller penetration and ductility. Therefore, the addition of carbon powder significantly increased the high-temperature performance of asphalt but had a negative impact on its low-temperature performance. This is because the chemical properties of carbon powder asphalt are very stable, and the addition of carbon powder changes the rheological properties of asphalt, which is then difficult to deform

**Table 5. Particle characteristic parameters of the two types of carbon powder.**

| Particle characteristic parameters | biochar powder made from straw | Commercial charcoal powder |
|---|---|---|
| Specific surface area | 829.9 $m^2$/kg | 599.3 $m^2$/kg |
| D [3,2] | 7.23 μm | 10.0 μm |
| D[4,3] | 19.80 μm | 25.10 μm |
| D(10) | 3.53 μm | 4.52 μm |
| D(50) | 9.68 μm | 20.4 0μm |
| D(90) | 35.00 μm | 53.6 0μm |

**Table 6. Technical performance of the asphalt under different biochar contents.**

| mass percentage of carbon powder content in asphalt binder/% | | 0 | 2 | 4 | 6 | 8 | 10 | 12 |
|---|---|---|---|---|---|---|---|---|
| biochar powder made from straw | Penetration (25˚C, 100 g, 5 s)/0.1 mm | 75.2 | 62.5 | 56.6 | 52.4 | 51.2 | 49.6 | 47.5 |
| | Softening point/˚C | 45.6 | 47.5 | 48.5 | 50.4 | 51.2 | 52.4 | 53.6 |
| | Ductility at 15˚C/(cm) | 110.4 | 56.6 | 32.8 | 24.5 | 21.8 | 18.2 | 14.6 |
| Commercial charcoal powder | Penetration (25˚C, 100 g, 5 s)/0.1 mm | 75.2 | 63.2 | 57.6 | 53.5 | 51.9 | 49.2 | 48.1 |
| | Softening point/˚C | 45.6 | 46.9 | 48.2 | 49.5 | 50.8 | 51.7 | 52.6 |
| | Ductility at 15˚C/(cm) | 110.4 | 57.2 | 33.5 | 25.3 | 22.4 | 20.1 | 16.5 |

in the high temperature season. Table 6 reveals that the technical properties of biochar and commercial charcoal modified asphalt with different carbon powder content are similar, indicating that the high temperature performance of the two modified asphalts is also similar.

In addition, each asphalt index changes rapidly in the early stage and tends to be smooth when the asphalt content exceeds 6%. Therefore, 6% biochar is recommended to produce asphalt.

In the construction of asphalt pavement, energy is consumed for heating the asphalt and aggregates and for mixing. Increasing the heating temperature of asphalt and the mixing time of the mixture will increase the energy consumption; moreover, increasing the mixing time will also reduce the mixing rate, affect construction progress, and increase construction costs.

The construction workability of asphalt pavement has a significant relationship with the viscosity–temperature characteristics of the asphalt binder [35,36]. China's technical code for highway asphalt pavement construction recommends the temperature corresponding to a viscosity of 0.17 ± 0.02 Pa·s as the mixing temperature of the asphalt mixture, and the temperature corresponding to a viscosity of 0.28 ± 0.03 Pa·s as the best compaction temperature of the

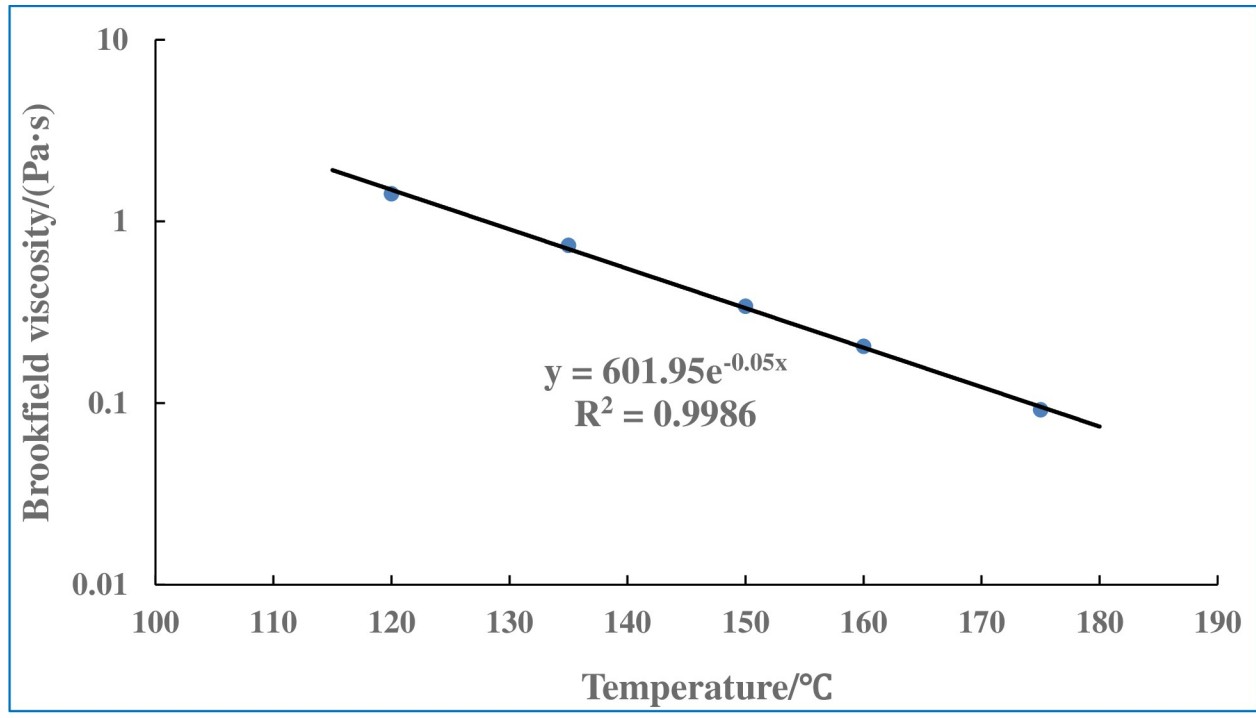

**Fig 4. Viscosity–temperature curve of the biochar-modified asphalt.**

asphalt mixture [37]. Therefore, in this study, the apparent viscosity of the asphalt binder with 6% biochar was measured at 135 and 175˚C [38]. Fig 4 shows plots of the viscosity–temperature curve of the biochar-modified asphalt in a semi-logarithmic scale.

From the figure, it can be seen that the temperature corresponding to a viscosity of 0.17 ± 0.02 and 0.28 ± 0.03 Pa·s is 161–166 and 151–156˚C, respectively. Therefore, the mixing and compaction temperatures of the biochar-modified asphalt mixture are 161–166 and 152–156˚C, respectively. These temperatures are slightly higher than the construction temperature of matrix asphalt No. 70.

## 6. Conclusions and recommendations

We confirmed that biochar can be prepared from crop straw and can be added into asphalt and found that the addition of crop straw biochar significantly improved the high-temperature performance of asphalt. Although the addition of crop straw reduces the low-temperature performance of asphalt, there is hardly any low-temperature road failure in high-temperature regions; thus, the low-temperature performance requirement of asphalt is very low. The addition of crop straw biochar was shown to increase the high-temperature performance of asphalt, which is beneficial to asphalt pavement in high-temperature regions. The crop straw biochar and commercial coal mainly consist of C, O, Si, and K, indicating that the biochar produced from crop straw has properties similar to commercial coal. Based on experimental results, we determined the appropriate amount of straw biochar powder to be used in asphalt to be 6%.

## Supporting information

**S1 Fig. EDS results of commercial charcoal powder.**
(TIF)

**S2 Fig. EDS results of biochar powder made from straw.**
(TIF)

## Author Contributions

**Conceptualization:** Wenli Zhang.

**Data curation:** Xinli Gan.

**Investigation:** Xinli Gan.

**Validation:** Wenli Zhang.

**Writing – original draft:** Xinli Gan.

**Writing – review & editing:** Xinli Gan, Wenli Zhang.

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
