## [Decision Letter · Decision Letter 0]

4 Sep 2020

PONE-D-20-24402

Application of biochar prepared from crop straw in asphalt modification

PLOS ONE

Dear Dr. Gan,

Thank you for submitting your manuscript to PLOS ONE. After careful consideration, we feel that it has merit but does not fully meet PLOS ONE’s publication criteria as it currently stands. Therefore, we invite you to submit a revised version of the manuscript that addresses the points raised during the review process.

We look forward to receiving your revised manuscript.

Kind regards,

Andrew R. Zimmerman, PhD

Academic Editor

PLOS ONE

Journal Requirements:

Additional Editor Comments (if provided):

Please pay special attention to the comments of Reviewer #1 who notes that much more discussion of the results is warranted and whole paper needs a thorough editing for proper grammar and English usage by a native English writer.

Reviewers' comments:

Reviewer's Responses to Questions

**Comments to the Author**

1. Is the manuscript technically sound, and do the data support the conclusions?

Reviewer #1: Partly

Reviewer #2: Partly

Reviewer #3: Partly

2. Has the statistical analysis been performed appropriately and rigorously? 

Reviewer #1: No

Reviewer #2: N/A

Reviewer #3: No

3. Have the authors made all data underlying the findings in their manuscript fully available?

Reviewer #1: Yes

Reviewer #2: Yes

Reviewer #3: Yes

4. Is the manuscript presented in an intelligible fashion and written in standard English?

Reviewer #1: Yes

Reviewer #2: Yes

Reviewer #3: Yes

5. Review Comments to the Author

Reviewer #1: Please see attached MS Word document. ----------------------------------------------------------------------------------------------------------------------------------------------------------------------

Reviewer #2: The study investigates the properties of asphalt binder modified with different dosages of biochar derived from crop straw. Some comments on the manuscript are as follows:

• The abstract states that “viscosity of the asphalt binder modified using different dosages of biochar powder were measured”. However, the viscosity results are presented for the binder with 6% biochar only. Please make suitable corrections to the abstract.

• No results of technical performance (penetration, softening point, ductility, viscosity) are presented for commercial charcoal modified asphalt binders. Please provide an objective statement, preferably in Section 1 of the manuscript.

• The authors have reviewed literature on the use of biochar for asphalt binder modification. I suggest the authors to also include the following relevant articles for an updated state-of-the-art on the subject:

(1) Wu, Y., Cao, P., Shi, F., Liu, K., Wang, X., Leng, Z., ... & Zhou, C. (2020). Modeling of the Complex Modulus of Asphalt Mastic with Biochar Filler Based on the Homogenization and Random Aggregate Distribution Methods. Advances in Materials Science and Engineering, 2020.

(2) Kumar, A., Choudhary, R., Narzari, R., Kataki, R., & Shukla, S. K. (2018). Evaluation of bio-asphalt binders modified with biochar: a pyrolysis by-product of Mesua ferrea seed cover waste. Cogent Engineering, 5(1), 1548534.

• Was an oxygen-deficit environment upheld during the generation of biochar? If yes, then it should be reported that the biochar was prepared under pyrolytic conditions.

• Did the authors observe any foaming of the asphalt binder on addition of biochar at the mixing temperatures?

• Placement under 120 °C for 6 hours will likely lead to an accelerated aging of the modified blend. Can authors justify this step? Were any arrangements made to ensure that the control binder also underwent a similar degree of aging before being subjected to testing?

• Authors are encouraged to discuss the implications of SEM and EDS results. How can the morphological features of biochar affect the asphalt-biochar interaction?

• Section 5.2: It is reported that “…biochar powder is more likely to agglomerate because of interfacial tension”. Since the interfacial tension was not measured in the study, please support this statement with appropriate reference(s).

• It is reported that biochar particles passing the 75-micron sieve were used in the study. However, Figure 5 shows some amount of biochar particles near the 1000-micron size. An explanation is needed for this observation. Could it be due to possible contamination by larger particles during the analysis?

• At several instances, the authors have emphasized that C content in biochar was found higher than charcoal. Please indicate what does it mean in physical terms: does it have a bearing on the asphalt-biochar blend properties?

• Some minor comments are:

o Replace ‘density’ with ‘ductility’ in Table 5 and also correct the units.

o Replace ‘brockfield’ with ‘Brookfield’ in Section 4.3.

o Replace ‘aggregation’ with ‘aggregates’ in Section 5.3.

Reviewer #3: The authors used a agriculture waste by product for enhancing the asphalt mixture properties. Although, they claimed that the main advantage of disposing of such waste material into environmental would cause environmental concern, still burning process (400 C) is used to produce bio-char material from it.

Since the authors did not include line number, I refer the comments through paragraph number.

1- Is not there any anther usage from strew waste materials? Recycling or in animal food science? It seems that the authors claimed that the only option, except land-filling, is to use it as asphalt material reinforcement agent! Please double check and verify your claim.

2- In figure 1, as far as I know, such straw is good for sheep grazing.

3- In table 1, specify the viscosity measurement temperature.

4- What is the advantage of providing elements of biochar straw? Once the authors have taken molecular dynamic simulation or investigate the physiochemical interaction between asphalt binders and modifier, providing such information would be beneficial.

5- The dosage shown in Table 5, is based on the total weight of binder? If so, please alluded it.

6- Based on old fashion test result, it is hard to believe that such agent would be beneficial for asphalt industry. To understand better, if you add any solid garbage in the asphalt binder, the same trend would be perceived! The advanced rheological test result would help to distinguish the different and make it advantageous be cleared.

6. PLOS authors have the option to publish the peer review history of their article (what does this mean?). If published, this will include your full peer review and any attached files.

Reviewer #1: No

Reviewer #2: No

Reviewer #3: No

---

## [Author Response · Author response to Decision Letter 0]

30 Nov 2020

We thank you and the reviewers for your thoughtful suggestions and insights. The manuscript has benefited from these insightful suggestions. I look forward to working with you and the reviewers to move this manuscript closer to publication in the PLOS ONE.

The manuscript has been rechecked and the necessary changes have been made in accordance with the reviewers’ suggestions. 

We revised the corresponding questions in Introduction, Lit Review, Materials and Methods, Results and Discussion in accordance with the reviewers’ comments, and rewritten the Conclusions.

---

## [Editor Report · Decision Letter 1]

2 Dec 2020

PONE-D-20-24402R1

Application of biochar prepared from crop straw in asphalt modification

PLOS ONE

Dear Dr. Gan,

Thank you for submitting your manuscript to PLOS ONE. After careful consideration, we feel that it has merit but does not fully meet PLOS ONE’s publication criteria as it currently stands. Therefore, we invite you to submit a revised version of the manuscript that addresses the points raised during the review process.

ACADEMIC EDITOR:

In your first revision, I only see responses to the first reviewer and not the second or third reviewer. Also, I did not see indication of the changes made in your 'track changes' revised manuscript submitted. Please be sure do have all this done before submitting again. Thanks

We look forward to receiving your revised manuscript.

Kind regards,

Andrew R. Zimmerman

Academic Editor

PLOS ONE

Additional Editor Comments (if provided):

Dear Dr. Gan,

In your first revision, I only see responses to the first reviewer and not the second or third reviewer. Also, I did not see indication of the changes made in your 'track changes' revised manuscript submitted. Please be sure do have all this done before submitting again. Thanks

---

## [Author Response · Author response to Decision Letter 1]

15 Dec 2020

We appreciated your letter regarding our manuscript (PONE-D-20-24402) titled “Application of Bio-char from Crop Straw in Asphalt Modification.” We are also extremely grateful to the reviewers for their comments on our manuscript. We carefully considered every comment and made thorough revision accordingly. In We have also addressed all their comments and answered all their questions in detail. If you have any other questions about this paper, I would appreciated it if you could let me know as soon as possible, so I can answered them.

Additive list

To reviewer #1:

Introduction :

Comment 1: First paragraph, 3rd line from bottom – I do not think “rational” is the word you want there. Suggest replacing with “sustainable.” 

Answer: Thanks for your very thoughtful suggestion. The manuscript has been thoroughly checked, and the mistakes have been fixed with the help of a native speaker. We have revised this sentence as follows: “Therefore, a sensible and sustainable way of using crop straw has become an important research and real-world application issue.”

Comment 2: First paragraph, sentence starting with “One possible method….” This is a sentence fragment/improper tenses/punctuation. It needs to be rewritten. 

Answer: Thanks for your very thoughtful suggestion. We have revised this sentence as follows: “Therefore, a sensible and sustainable way of using crop straw has become an important research and real-world application issue.”

Comment 3: Second paragraph, first sentence – remove the words “owing to repeated usage.” They are unnecessary, and the sentence should just start with “The performance of asphalt pavements….”

Answer: Thanks for your very thoughtful suggestion. We have revised this sentence as follows: “The performance of asphalt pavements inevitably declines under the combined action of load and the natural environment.”

Comment 4: Second paragraph, sentence that starts with “Under” – the word “owing” is used incorrectly. You mean “because of”

Answer: Thanks for your very thoughtful suggestion. We have revised this sentence as follows: “The performance of asphalt pavements inevitably declines under the combined action of load and the natural environment.”

Comment 5: Next sentence, remove “Being a cementitious material”

Answer: Thanks for your very thoughtful suggestion. We have revised this sentence as follows: “Asphalt, as a cementitious material, ages under the influence of light and temperature, which reduce its cementation ability and negatively affect the durability of asphalt pavement [7].” 

Comment 6: Sentence after that, remove the words “Therefore, owing” and replace with “Because of”

Answer: Thanks for your very thoughtful suggestion. We have revised this sentence as follows: “Owing to its high cost and environmental sensitivity, improving the durability of asphalt pavement has become the focus of road engineers.”

Comment 7: Third paragraph – extra “the” before adding. 

Answer: Thanks for your very thoughtful suggestion. We have added "the" before "adding"

Comment 8: Third paragraph, third line – remove the comma after mixtures. 

Answer: Thanks for your very thoughtful suggestion. We have removed the comma after “mixtures”, and revised this sentence as follows: “and aging resistance of asphalt and asphalt mixtures and does not adversely affect other asphalt pavement performance indicators [8, 9].”

Lit Review :

Comment 1: 4th line – remove the word “purposes” after “energy generation;” remove the dashes around the “such as” clause and replace with commas.

Answer: Thanks for your very thoughtful suggestion. We have revised this sentence as follows: “studied the economic benefit of the use of straw as a fuel, as raw material for papermaking, and as animal feed, compared to burning the straw in a field. Smuga-Kogut et al.”

Comment 2: Sentence that starts with S.M. Shafie – remove the comma after “logistical costs.” 

Answer: Thanks for your very thoughtful suggestion. We have removed the comma after“logistical costs.” 

Comment 3: Same sentence and next sentence – inconsistent formatting with your references. Above you were referring to your references by last name only. Then you did initials and last name. And here it is full first name and last name (Mingxing Sun). And then in the next sentence you are back to last name only (Nair). Which is correct? Be consistent and stick to it! 

 Answer: Thanks for your very thoughtful suggestion. We have revised the format of the references.

Comment 4: Next paragraph – same inconsistencies noted above (Zhao Sheng or just “Sheng” would probably be more correct); Renaldo Walters vs.”Walters.” Etc. 

 Answer: Thanks for your very thoughtful suggestion. We have revised the format of the references.

Comment 5: Sentence that starts with “Renaldo Walters” – you wrote 3 and 6%. The correct punctuation is “3% and 6%.” 

 Answer: Thanks for your very thoughtful suggestion. We have added "%" after "3"

Comment 6: Sentence starting with “FU Zhen” – this sentence does not make any sense unless you add the word “at” before the word “were.” (i.e. …asphalt that were conducted at….”

Answer: Thanks for your very thoughtful suggestion. We have revised this sentence as follows: “ Zhen et al. [24] prepared bio-char-modified asphalt via the high-speed shear method and conducted laboratory rolling thin film oven test and 60 °C dynamic viscosity tests of the bio-char-modified asphalt; they found that bio-char could strengthen the bond capability and resistance to flow deformation of the modified asphalt.”

Materials and Methods :

Comment 1: Sentence right after Fig. 2 – this should be in the past tense. I.e., the preparation process was as follows. 

Answer: Thanks for your very thoughtful suggestion. We have revised this sentence as follows: ”The preparation process was as follows”

Comment 2: Step 4 – the subject of this sentence is “the straws.” Therefore, the correct pronoun to refer to this subject is “they” and not “it.” 

Answer: Thanks for your very thoughtful suggestion. We have revised this sentence as follows: ” After cooling, the bio-char was ground to a powder using a grinder.”

Comment 3: Section 4.3 – What are the details associated with your physical property testing? Were there repetitions? If so, how many? Was a standard followed? If so, it must be referenced. 

Answer: Thanks for your very thoughtful suggestion. These tests followed the Standard Test Methods of Bitumen and Bituminous Mixtures for Highway Engineering (JTG E20-2011) of China. This has been specified in the manuscript. Following these standards, the ductility, softening point, and penetration tests were repeated three, two, and three times in each group, respectively. 

Results and Discussion :

Comment 1: Figs. 3 & 4 – I can see from the figure that the authors did not bother to make a “clean” document for submission and instead left the track changes on in MS Word. This is sloppy/one should expect better than this if this paper is truly ready for publication. 

Answer: Thanks for your very thoughtful suggestion. We are sorry for these problems. We have carefully proofread the manuscript and fixed all formatting issues.

Comment 2: Section 5.2 – third paragraph through the end of this section – I can clearly see the point the authors are making – bio-char has much smaller particles statistically than commercial-grade charcoal. However, I find myself wondering if this is the best way to present these data. Would it not be more efficient/concise to express Table 4 in graphical form similar to an aggregate particle distribution chart or a geotechnical soil particle chart? I would think this would make the data a little bit easier to read/interpret. In either case, remove the word “owing” in the 3rd to last line from the bottom of this section. Almost always, you can strike “owing” from your writing and either remove it entirely or replace it with the word “because.” 

Answer: Thanks for your very thoughtful suggestion. Because the particle size of carbon powder is very small (micron size), it is difficult to characterize it by the grading distribution diagram. Therefore, in this study, a laser particle size analysis test of the crop straw bio-char and commercial charcoal was conducted to evaluate the particle size characteristics of carbon powder. As suggested, we replaced the word “owing” with the word “because.” 

Comment 3: Section 5.3 – remove the word “moreover” from the sentence that starts with “As the biochar content increased…” Also add semicolon before “and the ductility”

Answer: Thanks for your very thoughtful suggestion. We have revised this sentence as follows: “It was found that, with the increase in bio-char content, the asphalt penetration decreased, the softening point increased, and the ductility decreased.”

Comment 4: Same sentence in Section 5.3 – you said that bio-char “significantly increased high-temperature performance.” Be careful with the word significantly unless you really mean it. It does not appear that there was any testing repetition, although based upon Section 4.3, it is very difficult to tell exactly what the authors did. If there were no repetitions, it would be very difficult to prove statistical significance. If the authors would be willing to couch their argument to something less definitive (i.e., remove the word “significant”) this would be fine as well. In either case, I would think a graphical presentation might be a better way to present these data as opposed to putting them in tabular form. Doing so would allow you to do much more interesting analysis like (for example) plotting a regression line through the data points. This does not prove statistical significance, but it does show a relationship between % biochar and physical properties and would be much more sophisticated than what is presented here. I just plotted the data myself and noted that there appears to be a linear relationship between softening and % bio-char; and apparent exponential relationships between penetration & density and % bio-char. Each of these relationships had relatively high R-squared values. 

Answer: Thanks for your very thoughtful suggestion. 

Many studies have shown that the softening point, penetration and ductility of asphalt are significantly related to the high temperature performance of asphalt. The better the high temperature performance of asphalt, the higher the softening point, the smaller penetration and ductility. The standard of these tests followed is the Standard Test Methods of Bitumen and Bituminous Mixtures for Highway Engineering (JTG E20-2011) of China. Each test was repeated in accordance with the requirements of the standard.

Conclusions :

Comment 1: The authors have dutifully summarized there work here. However, they have done almost nothing to connect the dots for the reader. In other words, what do these results imply for future asphalt mixtures? Might they be applicable to other types of asphalt with different aggregate?

Answer: Thanks for your very thoughtful suggestion. We rewrote the conclusions as follows: “Bio-char can be prepared from crop straw and can be added it into asphalt, especially in high-temperature areas. Although the biological addition of crop straw reduces the low-temperature performance of asphalt, there is hardly any low-temperature road failure in high-temperature regions; thus, the low-temperature performance requirement of asphalt is very low. The addition of crop straw bio-char increases the high-temperature performance of asphalt, which is beneficial to asphalt pavement in high-temperature regions. In addition, some studies have shown that the addition of commercial charcoal in the form of carbon powder can improve the high-temperature performance and fatigue performance of asphalt. The elements in both the crop straw bio-char and the commercial coal are mainly C, O, Si, and K. This indicates that the bio-char from crop straw has properties similar to the commercial coal. Based on experimental results, we determined the appropriate amount of straw bio-char powder to be used in asphalt to be 6 %, so the construction temperature should be increased appropriately.”

To reviewer #2:

Comment 1: The abstract states that “viscosity of the asphalt binder modified using different dosages of biochar powder were measured”. However, the viscosity results are presented for the binder with 6% biochar only. Please make suitable corrections to the abstract.

Answer: Thanks for your very thoughtful suggestion. We have revised this sentence in the abstract as follows: ”and the apparent viscosity of the asphalt binder with 6% bio-char was measured at 135 and 175 °C.”

Comment 2: No results of technical performance (penetration, softening point, ductility, viscosity) are presented for commercial charcoal modified asphalt binders. Please provide an objective statement, preferably in Section 1 of the manuscript.

Answer: Thanks for your very thoughtful suggestion. Previous studies on charcoal-modified asphalt used charcoal prepared from wood in the laboratory or commercial charcoal. We have specified this in Section 1.

Comment 3: The authors have reviewed literature on the use of biochar for asphalt binder modification. I suggest the authors to also include the following relevant articles for an updated state-of-the-art on the subject:

(1) Wu, Y., Cao, P., Shi, F., Liu, K., Wang, X., Leng, Z., ... & Zhou, C. (2020). Modeling of the Complex Modulus of Asphalt Mastic with Biochar Filler Based on the Homogenization and Random Aggregate Distribution Methods. Advances in Materials Science and Engineering, 2020.

(2) Kumar, A., Choudhary, R., Narzari, R., Kataki, R., & Shukla, S. K. (2018). Evaluation of bio-asphalt binders modified with biochar: a pyrolysis by-product of Mesua ferrea seed cover waste. Cogent Engineering, 5(1), 1548534.

Answer: Thanks for your very thoughtful suggestion. We have carefully read these two articles and cited them in our paper.

Comment 4: Was an oxygen-deficit environment upheld during the generation of biochar? If yes, then it should be reported that the biochar was prepared under pyrolytic conditions.

Answer: Thanks for your very thoughtful suggestion. Indeed, the biochar was prepared under pyrolytic conditions. The muffle furnaces was sealed to maintain an oxygen-deficit environment. We have specified this in the section "3. Materials". 

Comment 5: Did the authors observe any foaming of the asphalt binder on addition of biochar at the mixing temperatures?

Answer: Thanks for your very thoughtful suggestion. Yes, after the carbon powder is mixed into the asphalt, the air present in the pores of the biochar powder will enter the asphalt, causing the asphalt to produce some foam.

Comment 6: Placement under 120 °C for 6 hours will likely lead to an accelerated aging of the modified blend. Can authors justify this step? Were any arrangements made to ensure that the control binder also underwent a similar degree of aging before being subjected to testing?

Answer: Thanks for your very thoughtful suggestion. At higher temperature, the aging of asphalt is inevitable. In fact, any modification of the asphalt will inevitably lead to asphalt aging to a certain extent. Even asphalt stored in asphalt tanks above 120 ℃ (which is common before construction) ages. At present, there is no better way to prevent asphalt from aging.

Comment 7: Authors are encouraged to discuss the implications of SEM and EDS results. How can the morphological features of biochar affect the asphalt-biochar interaction?

Answer: Thanks for your very thoughtful suggestion. We have discussed some analysis results in the manuscript. When the powder is added into the asphalt binder, the asphalt will be wrapped on the surface of the powder particles, and interfacial tension will form under the combined action of intermolecular force between the two-phase, chemical bond, and mechanical locking forces.. Under the action of the interfacial tension, the asphalt will bond, thus reducing the plasticity of the asphalt. Finer carbon particles lead to larger surface area and stronger adsorption of asphalt.

Comment 8: Section 5.2: It is reported that “…biochar powder is more likely to agglomerate because of interfacial tension”. Since the interfacial tension was not measured in the study, please support this statement with appropriate reference(s).

Answer: Thanks for your very thoughtful suggestion. The expression of this sentence is indeed inaccurate and has been revised as follows: “Some studies have shown that smaller-sized particles (especially those smaller than 150 µm) form stronger agglomerates [29]. This is because the van der Waals force between the particles is strong when the particles are small, which induces the particles to agglomerate [30]. Because of its smaller particle size, the crop straw bio-char powder is more likely to agglomerate; hence, the crop straw bio-char-modified asphalt requires a longer shearing time during preparation. When the powder is added into the asphalt binder, the asphalt will be wrapped on the surface of the powder particles, and the interfacial tension will be induced under the combined action of the intermolecular force between the two-phase, chemical bond, and mechanical locking force. Under these effects, the plasticity of asphalt will be reduced, and deformation does not occur easily under the action of the force [31].”

Comment 9: It is reported that biochar particles passing the 75-micron sieve were used in the study. However, Figure 5 shows some amount of biochar particles near the 1000-micron size. An explanation is needed for this observation. Could it be due to possible contamination by larger particles during the analysis?

Answer: Thanks for your very thoughtful suggestion. Biochar particles passing the 75-micron sieve were used in the study. However, owing to the large number of carbon powder particles, it is inevitable that some particles greater than 75 microns will be mixed into the asphalt.

Comment 10: At several instances, the authors have emphasized that C content in biochar was found higher than charcoal. Please indicate what does it mean in physical terms: does it have a bearing on the asphalt-biochar blend properties?

Answer: Thanks for your very thoughtful suggestion. Carbon is stable in asphalt. The results show that the carbon content of bio-char is higher than that of charcoal, which indicates that the application of bio-char in asphalt modification is feasible. A higher carbon content will not adversely affect the performance of asphalt.

Comment 11: Some minor comments are:

o Replace ‘density’ with ‘ductility’ in Table 5 and also correct the units.

o Replace ‘brockfield’ with ‘Brookfield’ in Section 4.3.

o Replace ‘aggregation’ with ‘aggregates’ in Section 5.3.

Answer: Thanks for your very thoughtful suggestion. We have corrected these errors in the revised manuscript.

To reviewer #3:

Comment 1: Is not there any anther usage from strew waste materials? Recycling or in animal food science? It seems that the authors claimed that the only option, except land-filling, is to use it as asphalt material reinforcement agent! Please double check and verify your claim.

Answer: Thanks for your very thoughtful suggestion. At present, there are many ways to use crop straw apart from asphalt modification. We mentioned other methods to use crop straw in the introduction.

Comment 2: In figure 1, as far as I know, such straw is good for sheep grazing.

Answer: Thanks for your very thoughtful suggestion. China is a large agricultural country, so large quantities of straw are produced in the cultivation of crops. If sheep grazing is adopted, the treatment of these straws will be very slow. Moreover, many types of straws, such as corn straw, cannot be consumed by sheep. In addition, there are not enough sheep to consume all straw produced in crops.

Comment 3: In table 1, specify the viscosity measurement temperature.

Answer: Thanks for your very thoughtful suggestion. The viscosity was measured at 135 ℃; this has been specified in Table 1.

Comment 4: What is the advantage of providing elements of biochar straw? Once the authors have taken molecular dynamic simulation or investigate the physiochemical interaction between asphalt binders and modifier, providing such information would be beneficial.

Answer: Thanks for your very thoughtful suggestion. The elements in the crop straw bio-char are mainly C, O, Si, and K, which is consistent with the chemical composition of commercial charcoal. This indicates that the bio-char from crop straw has a similar effect to commercial charcoal. 

Comment 5: The dosage shown in Table 5, is based on the total weight of binder? If so, please alluded it.

Answer: Thanks for your very thoughtful suggestion. The dosage shown in Table 5 refers to the percentage by mass of asphalt binder. This has been specified in the manuscript.

Comment 6: Based on old fashion test result, it is hard to believe that such agent would be beneficial for asphalt industry. To understand better, if you add any solid garbage in the asphalt binder, the same trend would be perceived! The advanced rheological test result would help to distinguish the different and make it advantageous be cleared.

Answer: Thanks for your very thoughtful suggestion. This article aims to provide an application method for crop straw. Given the large amount of asphalt used for pavement paving, adding crop straws in the form of biochar into asphalt can consume a large amount of straws. Bio-char can be prepared from crop straw and can be added it into asphalt, especially in high-temperature areas. Although the biological addition of crop straw reduces the low-temperature performance of asphalt, there is hardly any low-temperature road failure in high-temperature regions; thus, the low-temperature performance requirement of asphalt is very low. The addition of crop straw bio-char increases the high-temperature performance of asphalt, which is beneficial to asphalt pavement in high-temperature regions.

---

## [Editor Report · Decision Letter 2]

18 Dec 2020

PONE-D-20-24402R2

Application of Bio-char from Crop Straw in Asphalt Modification

PLOS ONE

Dear Dr. Gan,

Thank you for submitting your manuscript to PLOS ONE. After careful consideration, we feel that it has merit but still does not fully meet PLOS ONE’s publication criteria as it currently stands. Therefore, we invite you to submit a revised version of the manuscript that addresses the points raised during the review process.

We look forward to receiving your revised manuscript.

Kind regards,

Andrew R. Zimmerman, PhD

Academic Editor

PLOS ONE

Additional Editor Comments:

I did not find that you fully addressed all the reviewers comments and I had a look myself and foufd further improvements needed as well. Track changes can be used when further editing the revised version.

-For this and ALL future submissions, line numbers need to be added to the manuscript and changes made need to be specific by line number

-Abstract needs to do a better job reporting novel results: specifically, HOW did addition of

crop straw bio-char significantly improve the high-temperature performance? How did performance compare with commercial charcoal?

- Change all ‘bio-char’ to ‘biochar’

-Could use more editing from native speaker. It is hard to believe that this was done given the awkwardness of language throughout the text. For example, tense use must made be consistent throughout manuscript

-For goodness sake, please stop answering every reviewer comment with: “ Thanks for your very thoughtful suggestion”

-‘ et al.’ cannot be used in the author lists in the bibliography

-Comments made by reviewers, should not only be responded to in the text (not just in comments back to reviewers). For example, when a review asked “Figure 5 shows some amount of biochar particles near the 1000-micron size.”, response/explanation should be made in the manuscript

-statistical significance needs to be considered in the text. For example, it the C content of the biochar significantly different from that of charcoal? Are you presenting averages? Need std. dev. This is true of ALL your parameters examined. Also, correct significant figures need to be correct and consistent

-Your first paragraph needs improvement. Incineration is NOT a straw production method. Also, incineration is certainly not the most common use of straw. How much straw is actually incinerated?

Related to this, I and reviewer 3 think you are mistakenly representing straw as something that needs to be gotten rid of. Is this the case? We doubt it. Please address in the conclusion, whether, given the minor benefits, this is really the best use of straw. Perhaps other asphalt additives that are truly waste would be better put to use here.

-Delete figure 1 and 2. Everyone knows what straw and charcoal looks like.

- Figures need to be numbered and ordered consecutively

-Fig. 6 has only two data points. This cannot make a correlation. It should be deleted along with all mention of a linear relationship

-Caption on figure 4 insufficiently describes the image

-ALL your asphalt data tables (tabs. 1 and 5), need to compare need to compare the properties of asphalt, with that of asphalt with biochar and asphalt with commercial charcoal. Otherwise, there is little value in this research and I will have to reject the manuscript.

Reviewer #3 Comment 4: What is the advantage of providing elements of biochar straw? Once the authors have taken molecular dynamic simulation or investigate the physiochemical interaction between asphalt binders and modifier, providing such information would be beneficial.

Answer: Thanks for your very thoughtful suggestion. The elements in the crop straw bio-char are mainly C, O, Si, and K, which is consistent with the chemical composition of commercial charcoal. This indicates that the bio-char from crop straw has a similar effect to commercial charcoal.

Editor: I don’t find this answer sufficient. Elemental composition is not an advantage. The abstract states: “bio-char significantly improved the high-temperature PERFORMANCE of asphalt”. This need to be more clear in abstract and conclusion.

- The dosage shown in Table 5 refers to the percentage by mass of asphalt binder. – should be added to the Table

-The ‘binder’ needs to be specified in the manuscript.

---

## [Author Response · Author response to Decision Letter 2]

29 Jan 2021

I wish to re-submit the manuscript titled “Application of biochar from Crop Straw in Asphalt Modification” for publication in PLOS ONE. The manuscript ID is PONE-D-20-24402R2.

I thank you and the reviewers for your thoughtful suggestions and insights. The manuscript has benefited from these insightful suggestions. I look forward to working with you and the reviewers to move this manuscript closer to publication in PLOS ONE.

The manuscript has been rechecked and the necessary changes have been made in accordance with the reviewers’ suggestions. The responses to all comments have been prepared given in “Response to reviwers”. 

Thank you for your consideration. I look forward to hearing from you.

---

## [Editor Report · Decision Letter 3]

8 Feb 2021

Application of biochar from Crop Straw in Asphalt Modification

PONE-D-20-24402R3

Dear Dr. Gan,

We’re pleased to inform you that your manuscript has been judged scientifically suitable for publication and will be formally accepted for publication once it meets all outstanding technical requirements.

Kind regards,

Andrew R. Zimmerman, PhD

Academic Editor

PLOS ONE
---

## [Editor Report · Acceptance letter]

15 Feb 2021

PONE-D-20-24402R3 

Application of biochar from Crop Straw in Asphalt Modification 

Dear Dr. Gan:

I'm pleased to inform you that your manuscript has been deemed suitable for publication in PLOS ONE. Congratulations! Your manuscript is now with our production department. 

Kind regards, 

on behalf of

Dr. Andrew R. Zimmerman 

Academic Editor

PLOS ONE